# A Long-Term Follow-Up of Dental and Craniofacial Disturbances after Cancer Therapy in a Pediatric Rhabdomyosarcoma Patient: Case Report

**DOI:** 10.3390/ijerph182212158

**Published:** 2021-11-19

**Authors:** Pei-Ching Chang, Shiao-Yu Lin

**Affiliations:** 1Department of Pediatric Dentistry, Taoyuan Chang Gung Memorial Hospital, Taoyuan City 33305, Taiwan; 2Department of Pediatric Dentistry, Lin-Kou Chang Gung Memorial Hospital, Taoyuan City 33305, Taiwan; lin321@adm.cgmh.org.tw

**Keywords:** dental disturbance, craniofacial disturbance, chemoradiation therapy, rhabdomyosarcoma

## Abstract

Rhabdomyosarcoma (RMS) is the most common soft tissue sarcoma in children and adolescents. A boy aged seven years and five months was diagnosed with stage three group III embryonal parameningeal RMS with intracranial extension. He received chemotherapy for 23 weeks in combination with localized radiotherapy during the inductive phase of nine weeks (a total tumor dose of 5040 cGy). Three months later, he was referred to the department of pediatric dentistry for radiation-induced caries, the treatment of which was later terminated because of severe trismus and radiation-induced oropharyngeal mucositis. Three years later, the patient returned for the fitting of a prosthesis because of mastication problems. The dental treatments performed included: extraction, banding, composite resin restorations, root canal fillings, and stainless steel crown fabrication. An interim denture was fitted due to the poor retention of the fixed prosthesis. As the patient grew older, they developed facial asymmetry as a result of the prominent atrophy of their right cheek. By the age of 32, they had lost multiple teeth and exhibited severe facial deformity. Therefore, it is essential not only to involve a multidisciplinary medical team before, during, and after cancer therapy, but also to initiate long-term follow-ups given the potential effects of late sequelae after chemoradiation in multiple developmental areas.

## 1. Introduction

Rhabdomyosarcoma (RMS), which is composed of neoplastic mesenchymal cells with varying degrees of striated muscle cell differentiation, is an aggressive malignant tumor and the most common soft tissue sarcoma in children and adolescents [1]. The incidence rates of RMS are 3.5% among children younger than 14 and 2% in adolescents aged 15 to 19 [2]. The rate of occurrence between males and females is 1.3–1.5 to 1 [3]. Primary rhabdomyosarcoma in the head and neck accounts for approximately 35% of cases in children, including lesions in parameningeal, orbital, and superficial areas [4]. The majority of pediatric patients suffer from Group-II and Group-III rhabdomyosarcoma with relatively favorable outcomes [5]. The survival rate has improved dramatically over the years [6], thanks to a multimodal regimen consisting of chemotherapy, surgery (for local control), and the selective application of adjuvant radiotherapy (RT) [7]. Chemoradiation, in particular, has been a key treatment strategy in the context of pediatric oncology, as pediatric patients may suffer the loss of oral functions, cosmetic compromise, and psychological distress if they undergo surgeries. However, chemoradiation can also impact the growth of soft and hard tissues in the affected areas, and, in the case of the head and face, cause facial and dental abnormalities. These anomalies become increasingly pronounced as patients grow older and can affect their quality of life. This report details the long-term dental and craniofacial disturbances in a pediatric RMS patient after this cancer therapy based on a follow-up that spanned 25 years.

## 2. Case Report

A boy aged seven years and five months and suffering from a protruding right eyeball, watering eyes, and sinusitis received a definitive diagnosis of stage three group III embryonal parameningeal RMS with intracranial extension. The large and heterogeneous tumor occupied the entire retrobulbar space and damaged the adjacent bones of the temporal floor, as well as the medial and inferior walls of the right orbital fossa. The lesions in the right parapharyngeal space extended to the maxillary sinus, nasal cavity, paranasal sinuses, ethmoid sinus, nasopharynx, sphenoid sinus, and the right orbital fossa. The intracranial tumor extended to the right para-cavernous sinus and the middle cranial fossa. The patient received chemotherapy (cisplatin, endoxan, adriamycin, oncovin, etoposide, vincristine, and actinomycin-D) for a period of 23 weeks commencing when the patient was seven years and five months old. This was combined with localized RT during the inductive phase of nine weeks. The radiation field included the tumor bed with a safe margin of at least 2 cm, the basal skull meninges, and the upper neck lymph node. These areas were irradiated at the Pediatric-Hematology Oncology Department using six MV X rays via one anterior portal and two lateral opposed portals with a total dose of 5040 cGy. Three months later, the patient was referred to the Department of Pediatric Dentistry for radiation-induced caries. However, the treatment was later terminated because of open-mouth limitation due to trismus and severely painful radiation-induced oropharyngeal mucositis.

Three years later, the patient returned for the fitting of a prosthesis because of mastication problems. The following were found in his primary and developing permanent dentitions: multiple caries and residual roots, crown opacity, hypoplastic teeth with apical lesions, foreshortened and blunted roots, incomplete root development, a prematurely closed root apex, and V-shaped roots (Figure 1a,b).

The objective of the dental treatment at the time, therefore, was to preserve the existing dentition for masticatory efficiency and to improve the esthetics of the patient’s teeth. Patient. The treatments performed included: extraction, banding, composite resin restorations, root canal fillings, and stainless steel crown fabrication. An interim denture was provided due to the poor retention of the provisional fixed restorations (Figure 2a). Six months later, following root canal fillings, all the new caries were restored, and the provisional prosthesis was fabricated (Figure 2b). Oral hygiene was also encouraged, in particular flossing, brushing, and the application of topical fluoride, during each three-month follow-up.

As the patient grew older, he developed facial asymmetry, triggered by the prominent atrophy of his right cheek. His chin tilted toward the right side of his face, creating a concave profile (Figure 3). During the six years after the cancer therapy, the maxilla changed little, whereas the mandible rotated counterclockwise (Figure 4). Compared with the age-appropriate norm, the cephalometric tracings taken at the ages of 11 and 17 showed limited craniofacial development (Table 1).

The orthodontic diagnosis included skeletal Class III malocclusion and dental Class III malocclusion with mandibular prognathism. However, orthodontic treatment coupled with orthognathic surgery was not recommended for this patient because of his susceptibility to pathological mobility and further root resorption due to atypical root morphology at multiple locations. At the age of 32, 25 years after the initial cancer therapy, the patient had lost multiple teeth and suffered severe facial deformity (Figure 5).

## 3. Discussion

Chemotherapy with definitive local RT can successfully treat RMS [8]. Approximately 80% of children diagnosed with RMS will survive after multimodal treatment, improved supportive care, and the refinement of tumor grouping and staging [9]. However, multimodal treatment has also been associated with dental and/or craniofacial complications as well as developmental defects among childhood cancer survivors (CCS) [10]. High-dose radiation and chemotherapy can each provoke dental and craniofacial complications, including mucositis, xerostomia, loss of taste, difficulty in swallowing, loss of appetite, nausea, malaise, weight loss [11], opportunistic infection, trismus, erythema, impaired dental and osseous development [12], and osteoradionecrosis of the mandible due to high bone density and low vascularity [11]. The most common developmental defects associated with chemoradiation therapy are: decreased growth (49%), facial asymmetry (37%), dental abnormalities (30%), visual (20%) and auditory impairment (18%), dysfunction and cognitive problems (17%), endocrine disorder (<10%), and a second malignancy (2%) [5].

Chemotherapeutic agents may change the spectrum of bacteria colonizing the oral cavity, favoring caries-related microflora [13]. Chemotherapy-induced xerostomia reduces the salivary flow and promotes a more acidic pH environment conducive to dental caries [14]. Chemotherapeutic agents can also affect mature secretory odontoblasts and ameloblasts and induce qualitative and quantitative changes in dental tissues. The interference with odontoblast microtubules disrupts the formation of collagen fibril and the secretion of dentin matrix, resulting in short and thin tapered roots, and the inhibition of odontogenesis and eruption [15]. Agents such as vincristine, vinblastine, and cyclophosphamide can disrupt the calcium transport mechanism of ameloblast microtubules, leading to hypomineralized enamel defects [16]. Chemotherapy administered during childhood has been associated with high incidences of dental defects, as immature teeth are at greater risk of dental abnormalities than fully developed ones [17]. Given the short half-life of chemotherapeutic agents, dental defects are usually localized as a result of the transient change in the odontoblast function rather than odontoblast cell death. Nevertheless, intensive and repeated chemotherapy during the period when hard tissues are starting to form may still cause tooth agenesis [12].

Radiation interferes with odontogenesis by directly inhibiting the mitotic activity of odontoblasts. The rapidly dividing presecretory odontoblasts in children are particularly vulnerable to the effects of radiation [18]. In contrast, radiation affects amelogenesis indirectly by inducing the formation of osteodentin, which replaces normal dentin [19]. Osteodentin reduces phosphorylated phosphoprotein, which inhibits the nucleation of enamel crystals and leads to deficient enamel mineralization [20]. The following radiation-induced dental disturbances are thus commonly seen: enamel hypoplasia and hypomineralization [17], microdontia and agenesis [21], aplasia, arrested root development with premature apical closure, foreshortening and blunting of roots, and eruption problems [17]. Late dental defects can occur with a radiation dose as low as 4 Gy. Mature ameloblasts can be damaged by 10 Gy. Tooth development can be halted by doses of up to 30 Gy [22]. Doses of over 60 Gy can cause irreversible injuries to bone cells and vascularity, as well as osteoradionecrosis and an unbalanced apposition–resorption pattern during tooth movement [23].

The patient in the current report regularly consumed unhealthy snacks and had a daily diet high in carbohydrates. The oral medications that he took contained a high amount of sucrose. He also experienced vomiting due to the chemotherapy. Hence, during all of the three-month follow-ups, oral hygiene was encouraged, especially flossing, brushing, and the application of topical fluoride.

Previous studies have demonstrated that intensive oral care can reduce the risk of moderate/severe mucositis without increasing the likelihood of septicemia and infections in the oral cavity [24,25]. Brushing using a soft toothbrush or an electric brush at least twice daily has been found to be the most efficient method of reducing the risk of significant bleeding and infections in the gingiva. Furthermore, the frequency of oral care should be increased in cases of tissue complications from the cancer treatment. Pain can be reduced by the application of anesthetics, analgesics, or mucosal coating agents, such as Benadryl, Kaopectate, milk of magnesia, Orabase, viscous Xylocaine, Oratect Gel, and systemic analgesics [11].

Due to the fact that used toothbrushes can quickly become oral bacteria colonies and potential sources of infection within two weeks [26], it is highly recommended not only to dry toothbrushes after use, but to soak them in chlorhexidine solution and to replace them regularly [27,28]. Extra soft brushes, sponge toothbrushes, toothettes, or disposable cleaning pads with foam rubber heads are all less effective than toothbrushes with soft nylon bristles when it comes to removing plaque and preventing dental caries [29,30]. These alternatives should only be used during severe mucositis when a patient cannot tolerate a regular toothbrush and flossing, or when their platelet count drops to 20,000 per cubic millimeter [11].

Flossing should be discouraged if the patient is not skilled at it because it can cause trauma and bleeding. Toothpicks and irrigators should also be avoided during neutropenia, as they may disrupt tissue integrity and create entry points for microbial colonization and bleeding [27]. Before the gingiva are healthy again, or mucositis has receded to the initial stage, patients with poor oral hygiene or periodontal disease can use chlorhexidine mouthwash daily to control early periodontal infections [26].

Periodontal infections are a concern because colonized organisms have been shown to cause bacteremia [24]. To keep the mouth moist and reduce pathogenic flora, an anti-plaque rinse such as isotonic saline or a 5% sodium bicarbonate solution is recommended. To control fungal activity, an antimicrobial agent such as oxytetracycline/amphotericin-B can be used [31]. Other products include artificial saliva, oral rinses, sprays, and fluoride in various forms, concentrations, and delivery methods. Patients who are at risk of caries induced by cancer therapies can continue the daily oral hygiene regimen while also initiating fluoride supplements and 1% neutral fluoride rinses or gels at least one week before the radiation therapy [32]. However, these measures should be performed judiciously if the patient is thrombocytopenic [26,33]. The frequency of professional applications of fluoride—fluoride varnish, fluoride gel, or fluoride foam— should be determined based on the patient’s risk profile [34,35]. High-strength fluoride toothpastes containing 5000 ppm of fluoride should be prescribed only to patients who can reliably spit out the toothpaste [34]. Fluoride trays can be offered to patients who can tolerate wearing them [36]. For those that are unable to excrete properly, products containing casein phosphopeptide and amorphous calcium phosphate (CPP-ACP)—such as MI Paste—can be used to maintain the surface mineralization of enamels [37]. Analgesic rinses, such as 2% viscous lidocaine, are also advisable for oral hygiene during mucositis. If the patient suffers from salivary dysfunction, moistening agents such as cholinergic receptor agonists (e.g., pilocarpine) are recommended [38,39].

Children who receive radiation therapy in the head and neck region may also develop trismus (painful spasms of the mastication muscles) [40,41]. Therefore, ideally, stretching exercises of the masticatory muscles through physical therapy should be initiated before the start, and continue beyond the completion, of the radiation therapy [42].

Children with intracranial tumors treated with RT in combination with multi-agent chemotherapy also suffer reduced vertical growth of facial structures [43]. Dahllof et al. reported that the values of all linear variables related to facial height were diminished significantly among bone marrow transplantation (BMT) patients treated with 10 Gy of total body irradiation (TBI). They found shorter anteroposterior facial dimensions and changes in the growth rotation of both the maxilla and mandible [44]. Impaired mandibular development has also been found in young leukemia patients after 24 Gy of cranial RT and chemotherapy [45]. According to Guyuron et al., the harmful dose could be as low as 400 cGy for soft tissue, and 3000 cGy for hard tissue [46]. The patient in the current case received localized RT with a total tumor dose of 5040 cGy, which was large enough to interfere with both the soft and hard tissue growth in his craniofacial structure. Mirroring the findings from previous studies [44], the patient’s limited craniofacial development became even more pronounced as he grew older, evidenced in his cephalometric measurements. Facial deformities and malocclusion also made him very depressed. This resonates with Pruzinsky’s observation that individuals with severe craniofacial deformities are at risk of experiencing social and psychological stress [47].

Approximately 77–100% of survivors who have undertaken head and neck radiation therapy report mild to severe radiation damages to soft tissues and bones [43,48]. The most pronounced complications of radiotherapy include hypovascularity and cytotoxic effects on epiphyseal chondrocytes [49,50]. Chemotherapy and RT also have the most notable effects during puberty [44,45,51]. Such impacts on the thyroid gland and pituitary axis can affect the patient’s overall growth [52].

In general, orthodontic treatment may start or resume after the completion of all cancer therapies and a disease-free survival of at least two years. By that time, the risk of relapse has decreased and the patient is no longer taking immunosuppressive drugs [53]. However, when treating long-term pediatric cancer survivors, orthodontists need to be mindful of the potential adverse clinical implications and risks of complications that may be triggered by the treatment administered. For example, the orthodontic appliances used may inflict stress to the oral mucosa and impair the membrane’s regenerative capability, leading to ulcerations [54]. To minimize risks, non-irritating orthodontic appliances should also be considered, such as nickel-free brackets (comprised of stainless steel with manganese or with a low nickel content of less than 5%) which reduce free radicals released from the stainless steel [54]. For nickel-sensitive patients, titanium brackets should be the first choice, as these are more resistant to corrosions and do not release nickel into the oral cavity [55].

In addition, a low force of 20 to 150 g per tooth helps to prevent unpredictable root resorption. [56]. The duration of treatment should be kept short. Research has shown that a pause of two to three months after six months of active treatment could reduce the number of patients experiencing advanced root resorptions [57]. In general, when treating patients at risk, it is recommended that periapical radiographs be taken after 6 months of active treatment. If the film reveals the signs of resorption, orthodontic treatments should be discontinued for three months [57].

Dahllof et al. have summarized the following strategies for providing orthodontic care to patients with dental sequelae: using appliances that minimize the risk of root resorption, using lighter forces, terminating treatments earlier than normal, choosing the simplest treatment method, and not treating the lower jaw [58]. However, late dental defects such as hypodontia and root stunting may preclude adequate orthodontic anchorage [59], placing teeth at an additional risk of the damaging effects of gingival and periodontal diseases [60]. This is the reason why orthodontic treatments combined with orthognathic surgeries were not recommended for the patient in the current study, as he had missed his follow-ups from the age of 17 and had only returned for further dental help till the age of 25.

RMS usually occurs in young children, and RT at a young age can lead to abnormal growth and functions of musculoskeletal tissues such as facial disfigurements (incidence rate: 35–77%) [61]. The experience of childhood cancer may also have a long-term impact on survivors’ psychological states and further hinder their post-treatment adjustment and affect their quality of life. As more patients survive cancer for longer periods of time, potentially serious complications of cancer therapies also begin to emerge [62]. Hence, as the prognosis of childhood cancer continues to improve, more attention should be directed towards the long-term side effects of treatment protocols, so as to improve patients’ quality of life and reduce the overall cost of care. General and pediatric dentists should also educate themselves in late complications, aiming for early diagnosis and the delivery of proper dental care for this growing population of patients.

## 4. Conclusions

It is essential to involve a multidisciplinary team of medical professionals in patient care before, during, and after cancer therapies. This multidisciplinary team may consist of oncologists, pediatric dentists, and other health care professionals who can collectively determine what combination of therapeutic modalities will deliver the optimal cost–benefit ratio, the best prognosis with minimal morbidity, and the best quality of life for patients. Professional oral and dental supervision are critical components of patient-centered care during pediatric cancer therapies [42]. A comprehensive oral care plan should include pretreatment evaluation and information collection, instructions for oral care, treatment of dental and periodontal diseases, and supportive follow-up care. Patients undergoing either radiation therapy or chemotherapy must understand that good oral hygiene is essential in reducing the chances of oral complications from their cancer treatments. The most important factor in ensuring a healthy oral cavity during cancer therapies is patient compliance. Educating caretakers as well as patients about the importance of oral care is also vital in minimizing discomfort and maximizing the chances of successful outcomes. Furthermore, late sequelae after the chemoradiation therapy may affect a number of developmental areas in pediatric patients: stature growth, cosmetic appearance, dental and maxillofacial abnormalities, and special senses. Long-term follow-ups, which focus on detection, prevention, and remediation of these complications, are thus needed for all patients.

## Figures and Tables

**Figure 1 ijerph-18-12158-f001:**
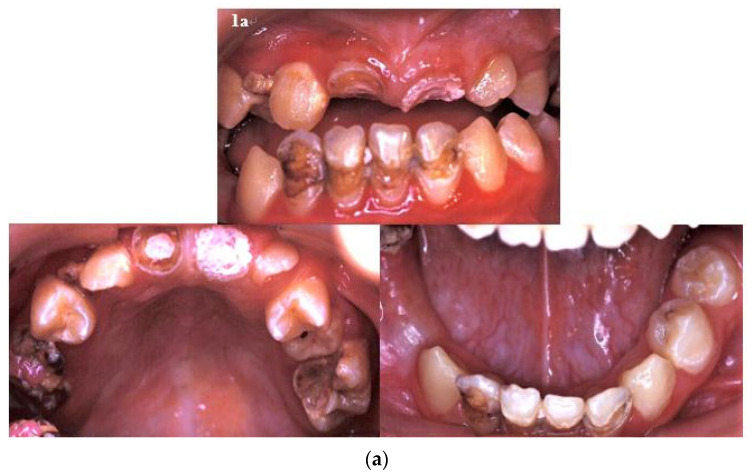
(**a**) There were multiple caries and residual roots, crown opacity, and hypoplastic teeth in the patient’s primary and developing permanent dentitions. (**b**) There were many deep caries with apical lesions, foreshortened and blunted roots, incomplete root development, prematurely closed root apex, and V-shaped roots in the developing dentition.

**Figure 2 ijerph-18-12158-f002:**
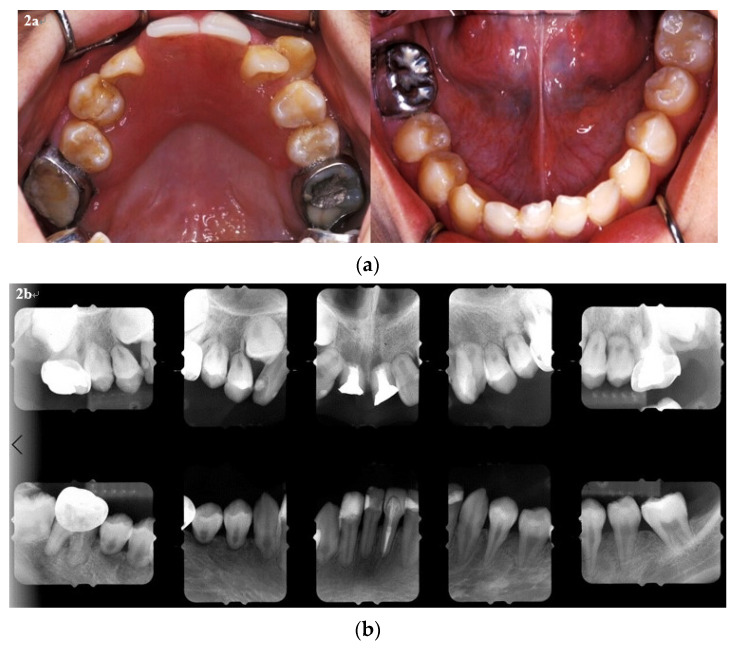
(**a**) The dental treatments performed included extraction, composite restorations, root canal fillings, stainless steel crown fabrication, banding, provisional restorations, and an interim denture. (**b**) Following root canal fillings, all new caries were restored, and the provisional prosthesis was fabricated.

**Figure 3 ijerph-18-12158-f003:**
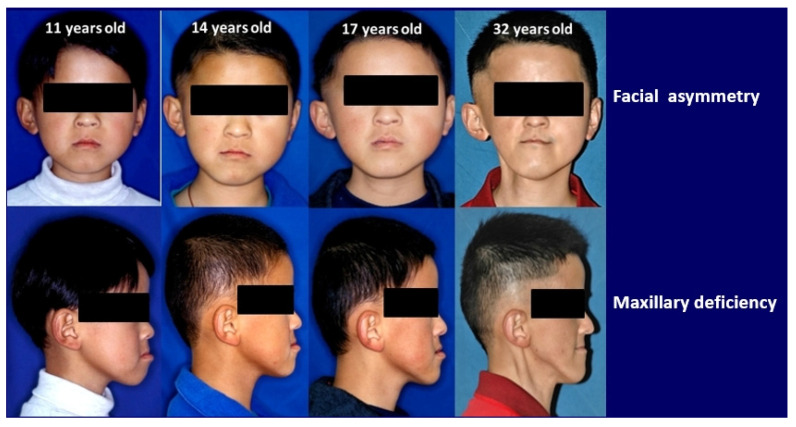
Long-term facial asymmetry as a result of pediatric rhabdomyosarcoma (RMS) cancer therapy.

**Figure 4 ijerph-18-12158-f004:**
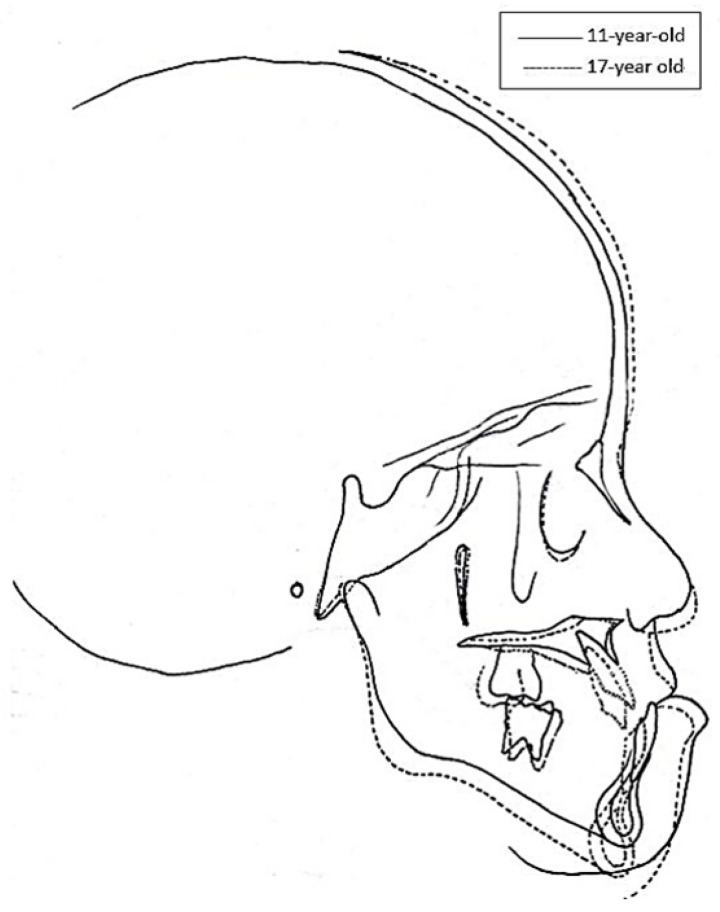
Superimposition of cephalometric tracings taken at the ages of 11 and 17 years.

**Figure 5 ijerph-18-12158-f005:**
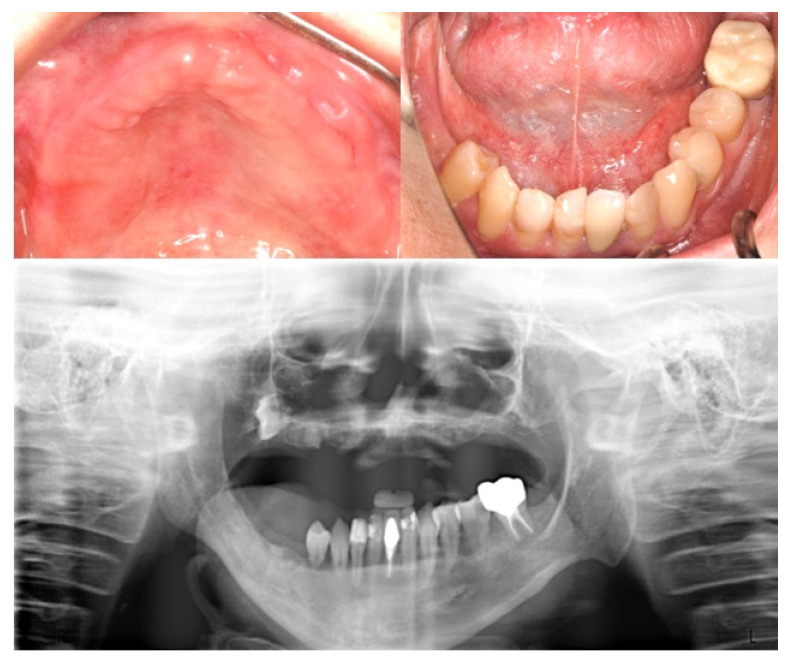
The patient lost multiple teeth and suffered severe facial deformity 25 years after the pediatric rhabdomyosarcoma (RMS) cancer therapy.

**Table 1 ijerph-18-12158-t001:** Cephalometric tracing measurements.

		11 y/o		17 y/o
	P’t	Mean ± SD	P’t	Mean ± SD
SN	58	67.67 ± 2.82	58	73.37 ± 3.64
SN- FH	5.5	7.36 ± 2.67	6.5	6.04 ± 3.32
SN- MP	41	33.10 ± 4.69	39	26.38 ± 6.34
SNA	80	82.00 ± 3.24	80	83.99 ± 4.20
SNB	82	77.60 ± 3.21	81	81.76 ± 4.03
ANB	−2	4.40 ± 1.56	−1	2.24 ± 2.13
A-Nv	−5	−0.81 ± 2.36	−3.5	0.02 ± 3.59
B-Nv	−4	−8.91 ± 4.07	−4	−4.24 ± 5.12
S-Go (PFH)	64	75.18 ± 3.99	72	94.53 ± 6.89
N- Me (AFH)	108	117.41 ± 4.84	115	134.09 ± 5.16
N-A (UFH)	51	59.27 ± 3.12	52	67.66 ± 2.79
A-Me (LFH)	57	59.17 ± 3.52	63	66.99 ± 4.13
PFH/AFH %	59.26%	64.11 ± 3.78	62.61%	70.61 ± 5.92
UFH/LFH%	89.47%	100.50 ± 7.64	82.54%	101.28 ± 6.65
Go-Gn	67	71.62 ± 3.09	67	85.36 ± 4.65
Ar-A (mm)	69	83.35 ± 3.41	69	94.29 ± 4.82
Ar- Gn (mm)	99	101.73 ± 3.69	104	122.18 ± 6.61
Ar-Go (mm)	39	42.97 ± 3.00	46	58.15 ± 6.15
Ar-Go-Gn	137	123.35 ± 3.87	133	115.96 ± 8.31
U1-SN	111	105.13 ± 5.92	101	110.46 ± 6.35
L1-MP	77	98.31 ± 5.67	75	101.56 ± 8.73

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
