# Peer review of "A Long-Term Follow-Up of Dental and Craniofacial Disturbances after Cancer Therapy in a Pediatric Rhabdomyosarcoma Patient: Case Report"

_ijerph, 2021, doi:10.3390/ijerph182212158_

Round 1

Reviewer 1 Report

This is a very interesting and needed article about a concerning cancer common in children.

I just suggest some improvements:

-Introduction: talking about most frequent localizations in head and neck region of Rhabdomyosarcoma could be interesting, and also the most common grade.

-Case Report and Discussion: talking about the dental preventive measurements used (as topical fluor application) or not used during chemotherapy and/or radiotherapy, could be interesting. Also if the patient did not recieved any of them, and explanation about the common preventive measurementes and the difficulties and reasons for not applying them, is suggested.

-Throughout the text: the term "radiotherapy" is used constantly, although only at line 214 is defined and acronyms are used. Please, explain the acronyms for the benning and after de explanation use always RT to refer to the term.

Author Response

Comment: Introduction: talking about most frequent localizations in head and neck region of Rhabdomyosarcoma could be interesting, and also the most common grade. 

Response: Thank you for the comment. As suggested by Reviewer, the most frequent rhabdomyosarcoma in head and neck as well the most common grades have been added to Introduction (see p. 1).

Comment: Case Report and Discussion: talking about the dental preventive measurements used (as topical fluor application) or not used during chemotherapy and/or radiotherapy, could be interesting. Also if the patient did not receive any of them, and explanation about the common preventive measurements and the difficulties and reasons for not applying them, is suggested.

Response: Thank you for the comment. As suggested by Reviewer, the common preventive measures and those used in our study are now elaborated on in Discussion (see Paragraphs 3-5, p. 7; Paragraphs 1-4, p. 8).

Comment: Throughout the text: the term "radiotherapy" is used constantly, although only at line 214 is defined and acronyms are used. Please, explain the acronyms for the beginning and after de explanation use always RT to refer to the term.

Response: Thank you for the comment. "Radiotherapy" is now spelled out in Introduction (p. 1), with its mentions in the rest of the manuscript abbreviated as "RT".

Reviewer 2 Report

The Authors presented a single patient case study of 25 years follow-up after 23 weeks of oncological therapies. The described orthodontic problems, although supported by statistical data (Hicks J.; Flaitz C. Rhabdomyosarcoma of the head and neck in children. Oral Oncol. 2002, 38, 450-459), can be only partially attributed to the the oncological therapy and can strongly depend on the orthodontic therapy choice (line 150 "the orthodontic treatment coupled with orthognathic surgery was not recommended to the patient"). 

The Authors lack in the description of similar patients case studies that were treated with different orthodontic therapy approaches. Without a comparative analysis and discussion this paper is not significantly scientifically exhaustive.

Author Response

Comment: The Authors presented a single patient case study of 25 years follow-up after 23 weeks of oncological therapies. The described orthodontic problems, although supported by statistical data (Hicks J.; Flaitz C. Rhabdomyosarcoma of the head and neck in children. Oral Oncol. 2002, 38, 450-459), can be only partially attributed to the oncological therapy and can strongly depend on the orthodontic therapy choice (line 150 "the orthodontic treatment coupled with orthognathic surgery was not recommended to the patient").

The Authors lack in the description of similar patients case studies that were treated with different orthodontic therapy approaches. Without a comparative analysis and discussion this paper is not significantly scientifically exhaustive.

Response:  Thank you for the comment. As suggested by Reviewer, orthodontic treatments among similar patients in other studies are now added to Discussion and analyzed in greater depth. The reasons why orthodontic treatments combined with orthognathic surgeries were not recommended to the patient in our report were also expounded (see Paragraphs 2-5, p. 9).

Reviewer 3 Report

Minor comments:

In the discussion chapter, the authors focus more on the effects of chemo- and radiotherapy on dental and craniofacial structures and not enough on treatment alternatives or possible solutions for the discussed patient or patients in general.

Author Response

Reviewer 3

Comment: In the discussion chapter, the authors focus more on the effects of chemo- and radiotherapy on dental and craniofacial structures and not enough on treatment alternatives or possible solutions for the discussed patient or patients in general.

Response:  Thank you for the comment. As suggested by Reviewer, treatments reported in other studies are now added to Discussion and addressed in greater depth. The reasons why orthodontic treatments combined with orthognathic surgeries were not recommended to the patient in our report were also expounded (see Paragraphs 2-5, p. 9).

Round 2

Reviewer 2 Report

The paper has been sufficiently revised